# Genome-Wide Identification of BTB Domain-Containing Gene Family in Grapevine (*Vitis vinifera* L.)

Nandni Goyal, Monika Bhuria, Deepika Verma, Naina Garewal and Kashmir Singh *

Department of Biotechnology, BMS Block I, Panjab University, Sector 25, Chandigarh 160014, India
* Correspondence: kashmirbio@pu.ac.in or kashmir123@gmail.com; Tel.: +91-172-2534085

**Abstract:** BTB (broad-complex, tram track and bric-a-brac) proteins have broad functions in different growth processes and biotic and abiotic stresses. However, the biological role of these proteins has not yet been explored in grapevine, which draws our attention towards the *BTB* gene family. Herein, we identified 69 *BTB* genes (*VvBTB*) in the *Vitis vinifera* genome and performed comprehensive in silico analysis. Phylogenetic analysis classified VvBTB proteins into five groups, and further domain analysis revealed the presence of other additional functional domains. The majority of BTB proteins were localized in the nucleus. We also performed differential expression analysis by harnessing RNA-seq data of 10 developmental stages and different biotic and abiotic stresses. Our findings revealed the plausible roles of the *BTB* gene family in developmental stages; Fifty *VvBTB* were differentially expressed at different developmental stages. In addition, 47 and 16 *VvBTB* were responsive towards abiotic and biotic stresses, respectively. Interestingly, 13 *VvBTB* genes exhibited differential expression in at least one of the developmental stages and biotic and abiotic stresses. Further, miRNA target prediction of 13 *VvBTB* genes revealed that vvi-miR482 targets VvBTB56, and multiple miRNAs, such as vvi-miR172, vvi-miR169 and vvi-miR399, target *VvBTB24*, which provides an insight into the essential role of the BTB family in the grapevine. Our study provides the first comprehensive analysis and essential information that can potentially be used for further functional investigation of *BTB* genes in this economically important fruit crop.

**Keywords:** *Vitis vinifera*; BTB; abiotic; biotic; development; miRNA



## 1. Introduction

BTB, also known as POZ (pox virus and zinc finger), is a highly conserved and interactive domain consisting of 115–120 amino acid residues frequently present at the N-terminus of proteins [1] whose function is not yet fully explored. This domain consists of five alpha-helices and three beta-sheets and acts as a protein–protein interaction module that exhibits both self-interactions and associations with other proteins [2]. BTB proteins are usually known to affect the expression of other genes, primarily due to the presence of DNA-binding domains along with the BTB domain [3]. Based on this, the BTB protein family has been divided into various subfamilies, such as BTB-BACK, MATH-BTB, BTB-ANK, BTB-only, BTB-NPH3, BTB-ZF, BTB-BACK-Kelch, BTB-DUF, etc. [2,4,5]. According to some recent reports, the BTB domain mediates the oligomerization of NPR1, highlighting its potential role in biotic stresses [6,7]. The primal recognition of BTB domain proteins was achieved in *Drosophila melanogaster* [8]. However, the excavation of this domain in plant genomes has been performed in recent years. In *Arabidopsis*, a total of 150 proteins are known to contain the BTB domain at the N and C terminal extension regions [9,10]. Genome-wide investigation studies of BTB proteins have been performed in few plant species till now, for, e.g., tomato (*Solanum lycopersicum*), rice (*Oryza sativa*) and sugar beet (*Beta vulgaris*) [11–13]. Furthermore, BTB proteins were also found to be involved in the developmental patterns of various plants. For instance, *Arabidopsis* BT (BTB AND TAZ) members exhibited functional redundancy and major roles in gametophyte development.

Likewise, the BT2 protein regulates plants' responses towards ABA and sugars by interacting with two global transcription factor group E proteins (GTE9 and GTE11), and BT1 and BT2 are also known to play crucial roles in nitrate responses by direct activation through NLP transcription activators [14–16]. In addition, the BT2 protein also regulates telomerase activity in vegetative organs and is suggested to be a component of several interconnected networks [17–20]. MdBT2 also delays leaf senescence by interacting with the MdbHLH93 protein in apple [21]. Furthermore, BTB proteins are known to regulate abiotic and biotic stresses in various plant species, such as *Arabidopsis*, tomato, sweet potato, rice, *Capsicum annum*, maize, sugar beet, apple, *Nicotiana benthamiana* and soyabean [11–13,22–26].

Being a perennial fruit crop model, grape can be used to provide a better understanding of different developmental stages, including berry development. However, grape productivity is greatly hampered by different abiotic and biotic stresses. The already available grape genome serves as a genetic resource for the genome-wide identification of important gene families [27]. Moreover, the function of *BTB* genes remains unexplored in grapevine, which is both a commercially and economically important fruit crop. Hence, we engaged our focus on a genome-wide investigation of the *BTB* gene family in this plant. Thus, in the present study, we retrieved RNA-seq data for the developmental stages and abiotic and biotic stresses of *V. vinifera* and predicted differentially expressed *BTB* genes in different conditions. Additionally, we investigated selected *BTB* genes as the targets of several development- and stress-related miRNAs of *V. vinifera*. Altogether, our study provides candidate *BTB* genes involved in developmental and stress-related pathways that can be genetically engineered to improve the internal mechanisms of grapevine varieties.

## 2. Materials and Methods

### 2.1. Mining of BTB Genes in V. vinifera

First, coding sequences (CDS) of *BTB* genes of *V. vinifera* were retrieved from the National Center for Biotechnology Information (NCBI) genome database (ftp://ftp.ncbi.nlm.nih.gov/genomes/Vitisvinifera/protein/(accessed on 12 April 2021)) for local database generation. To identify the BTB domain-containing sequences in the grape genome, known BTB domain-containing protein sequences of *P. trichocarpa*, *S. lycopersicon*, *A. thaliana*, *O. sativa*, *G. max* and *V. vinifera* were retrieved from Uniprot (https://www.uniprot.org/uniprot/?query (accessed on 12 May 2021)) to make a query file. The alignment of query file sequences against the above generated local database was performed using standalone tBLASTn with an e value of $1 \times 10^{-5}$ After the prediction of putative hits, further analysis of the sequences of candidate genes was performed using the Grape Genome Browser (12×) (http://www.genoscope.cns.fr/externe/GenomeBrowser/Vitis/ (accessed on 25 December 2021)), the Pfam database (https://pfam.xfam.org/ (accessed on 25 December 2021)), the Conserved Domains Database (CDD) (http://www.ncbi.nlm.nih.gov/Structure/cdd/wrpsb.cgi (accessed on 25 December 2021)), ScanProsite (https://prosite.expasy.org/scanprosite/ (accessed on 26 December 2021)) and SMART (Simple Modular Architecture Research Tool) (http://smart.embl-heidelberg.de/ (accessed on 26 December 2021)) [28–31]. Consequently, the identified proteins were confirmed for the presence of the BTB domain in their sequences.

### 2.2. Chromosomal Mapping and Gene Nomenclature

The chromosomal locations of *BTB* genes of *V. vinifera* were determined with the assistance of NCBI. Mapping was performed through MapInspect1.0 software (https://mapinspect.software.informer.com/ (accessed on 1 October 2022; Wageningen University, The Netherlands). For nomenclature, Vv (*V. vinifera*) was used as a prefix for *BTB* genes and numbered according to the position on the chromosome from top to bottom.

### 2.3. Phylogenetic Analysis

The multiple sequence alignment of protein sequences of identified BTB proteins was completed using the ClustalW program within MEGA7 (http://www.ebi.ac.uk/Tools/msa/clustalw2/ (accessed on 2 May 2022; Pennsylvania State University, USA) with the default parameters [32]. A phylogenetic tree was constructed in MEGA7 software by the maximum likelihood method with 1000 bootstrap replicates [33]. Visualization of the phylogenetic tree was achieved with iTOL v6 software, using the Newick format as input (https://itol.embl.de/ (accessed on 15 February 2022; EMBL, Germany) [34].

### 2.4. Gene Structure, Motif and Domain Analysis

The intron–exon location of *BTB* genes was displayed using the GSDS 2.0 server (http://gsds.cbi.pku.edu.cn/ (accessed on 18 February 2022) by aligning the genomic sequence with CDS [35]. Further verification was performed using Ensembl Plants Archive release 49 (http://plants.ensembl.org/Vitis_vinifera/Gene (accessed on 18 February 22).

Conserved motifs were identified using MEME-suite version 5.4.1 (https://meme-suite.org (accessed on 3 January 2023). A total of 20 motifs with a width of 20–200 were analyzed in BTB proteins. Motif analysis was performed using InterProScan. Domain analysis of each candidate protein was completed after confirmation from the NCBI-CDD database.

### 2.5. Physicochemical and Structural Analysis

The physico-chemical characteristics of BTB proteins were explored through the Prot-Param ExPasy tool (https://web.expasy.org/protparam/ (accessed on 3 May 2022). It deciphers the amino acid number, molecular weight, theoretical pI (isoelectric point), the aliphatic index and the instability index [36]. The subcellular localization was analyzed using CELLO v.2.5 (http://cello.life.nctu.edu.tw/ (accessed on 3 June 2022) and WoLF PSORT (https://wolfpsort.hgc.jp/ (accessed on 3 June 2022) [37,38]. The prediction of tertiary structure was achieved using I-TASSER v5.1 (Iterative Threading ASSEmbly Refinement) based on homology modeling (https://zhanglab.ccmb.med.umich.edu/I-TASSER/ (accessed on 15 March 2022), and further secondary conformations were shown using Endscript 2 v2.0.11 (https://endscript.ibcp.fr/ESPript/cgi-bin/ENDscript.cgi (accessed on 16 March 2022)) [39,40].

### 2.6. Transcriptomic Data Collection and Expression Profiling of BTB Genes

RNA-seq data of different biotic, abiotic and developmental conditions such as PM (leaves of 5-month-old potted plants inoculated with *Erysiphe necator* at 36 h post-inoculation), DM (leaf discs of glasshouse-grown vines inoculated with *Plasmopara viticola* at 24 hpi and 48 hpi), cold (leaves of well-grown potted plants exposed to 0 °C for 3, 12, 48 and 72 h), heat (deseeded berries at 5–6 and 12–14 weeks post-flowering exposed to 38 °C for 1 h) (PRJNA149155), drought (leaves of 9-week-old potted plants without watering at 2nd, 4th and 8th day), inflorescence (3, 5 and 7 days after 100% cap-fall), berry (veraison, intermediate and mature) and leaf (young, medium- and large-sized and mature) of *V. vinifera* were obtained from the NCBI Sequence Read Archive (SRA) (http://www.ncbi.nlm.nih.gov/sra (accessed on 4 April 2022) based on different studies [41–48]. The details of the RNA-seq data are included in Table S1. For further identification of potential candidates of *BTB* genes in *V. vinifera*, we performed expression profiling using the Trinity-V 2.05 package [49]. The expression levels in the form of FPKM (fragments per kilobase of transcript per million fragments mapped) values were quantified by RSEM (RNA-seq by Expectation-Maximization) software (accessed on 5 April 2022; University of Wisconsin-Madison, USA) as a part of the Trinity package (Broad Institute, USA and the Hebrew University of Jerusalem, Israel). Further, differentially expressed genes (DEGs) were analyzed. Subsequently, DEGs that displayed at least a 1.5-fold change in expression level with a significance score of 0.05 were selected for further analysis. The generation of heat maps and hierarchical clustering (HCL) were achieved using TBtools [50]. Heat maps were generated taking the $\log_2$ of gene expression values with default parameters.

### 2.7. Prediction of Cis-Regulatory Elements

The promoter sequences of differentially expressed *BTB* genes were investigated for the prediction of *cis*-regulatory elements. The genomic sequences (1.5 kb) upstream of the translation start site were used for promoter analysis, which was retrieved from the Grape Genome Browser. Both (+) and (−) regions of promoters were analyzed using the PlantCARE database (http://bioinformatics.psb.ugent.be/webtools/plantcare/html/ accessed on 20 April 2022) [51], and TBtools v1.098765 (South China Agricultural University, Guangzhou, China) was used for visualization [50].

### 2.8. miRNA Target Prediction

Interactions between grape-specific miRNAs and *BTB* genes were predicted using the plant small RNA Target analysis server (psRNATarget accessed on 24 April 2022) with the default parameters [52]. Further representation of target *BTB* genes with their specific miRNAs was achieved using Cytoscape 3.9 (Institute of Systems Biology, Seattle, USA) [53].

## 3. Results

### 3.1. Genome-Wide Identification and Chromosomal Distribution

The extensive tBLASTn search resulted in the identification of 69 putative *BTB* genes in the *V. vinifera* genome. Further validation was performed using different databases that confirmed the presence of the BTB domain. To investigate the chromosomal location of *BTB* genes, a location map of each chromosome was constructed (Figure 1). The 69 *BTB* genes were distributed on all the chromosomes (Chr) except 9 and 16. The highest number of *BTB* genes (8) were present on Chr 7 and 8, followed by five on Chr 2 and four on Chr 6, 10, 15, 18 and 19, respectively. The lowest number of *BTB* genes (1) were localized on Chr 11, 14 and 17. In addition, five *BTB* genes were located on Un Chr, and three genes were at random positions on Chr 3, 11 and 12. Overall, our mapping results suggested that *BTB* genes are broadly allocated in the *V. vinifera* genome.

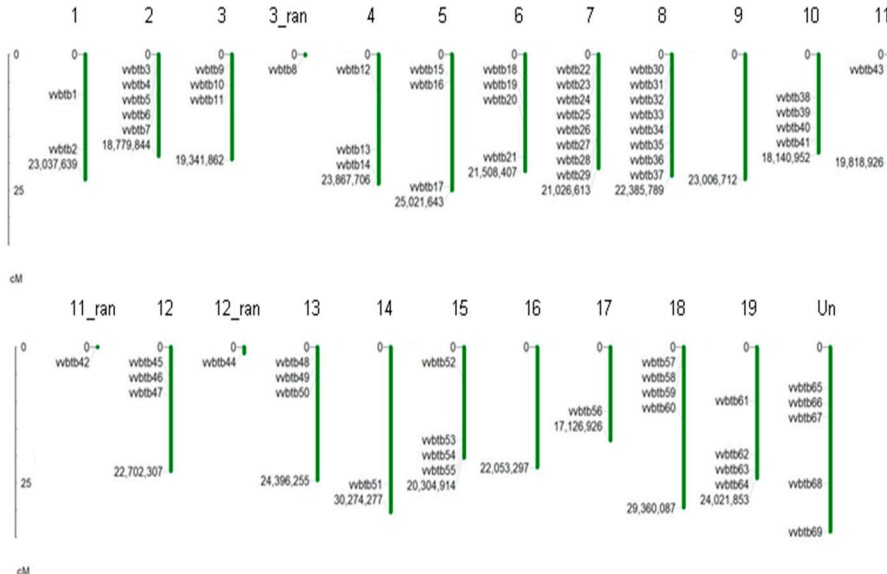

**Figure 1.** Chromosomal mapping of 69 *BTB* genes in the genome of *V. vinifera*. The 69 *BTB* genes are mapped onto grape chromosomes. Chromosome number is specified on the top of chromosome bars. The name and location of each gene is written on left side of each chromosome. The scale on the left side shows the length of chromosomes in cM (centimorgans). However, those members whose exact physical location was not known were represented as 3_ran, 11_ran and 12_ran.

### 3.2. Phylogenetic Analysis

The phylogenetic relationships among 69 BTB candidates were explored by the construction of a maximum likelihood tree (Figure 2), which divided BTB family proteins of grapevine into five major groups. Group I was the largest group, containing 30 BTB proteins. Similarly, groups II, III, IV and V consisted of 12, 2, 16 and 9 proteins, respectively. The proteins present in the same group exhibited similar protein sizes and intron–exon distribution at the genomic level.

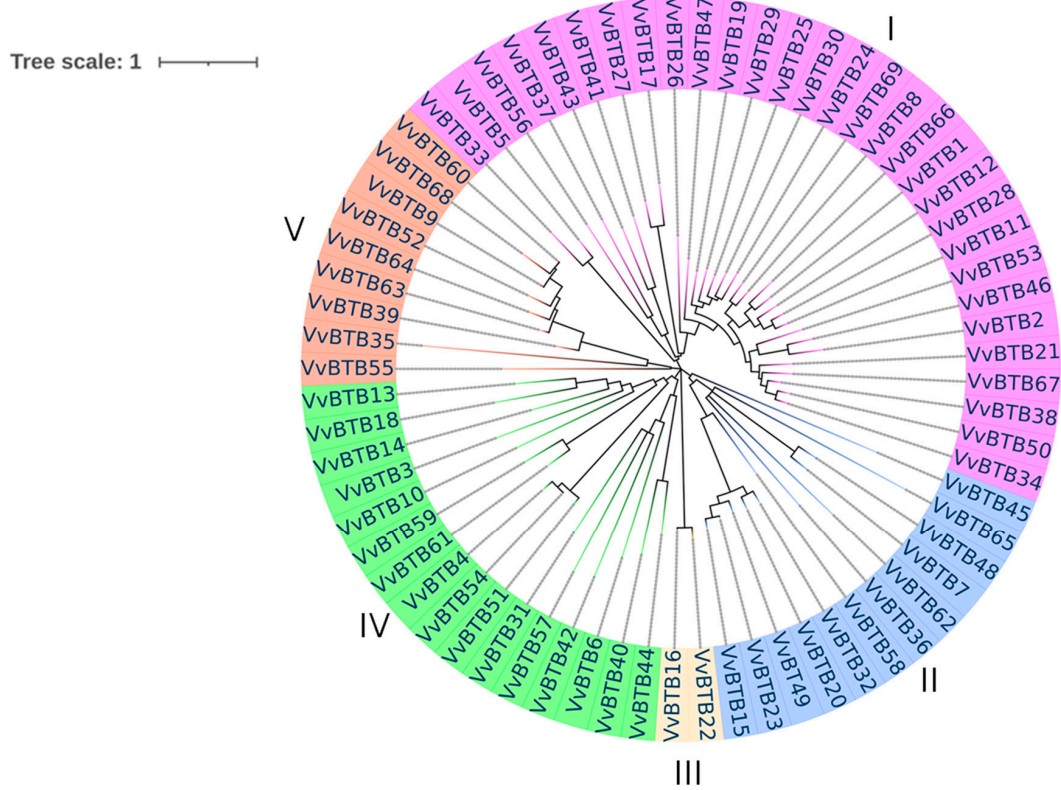

**Figure 2.** Phylogenetic tree of *BTB* gene family. The maximum likelihood tree was generated using protein sequences of 69 *BTB* genes of *V. vinifera*. Different colors show five different groups. Pink color: group I; blue color: group II; light-yellow color: group III; green color: group IV; red color: group V.

### 3.3. Gene Structure and Domain Analysis

As shown in Figure 3, intron–exon organizations of *BTB* genes in *V. vinifera* were displayed using the GSDS2.0 server. It was observed that groupwise structural similarities were present in *VvBTB* genes. The exon and intron numbers of *VvBTB* genes ranged from 1 to 18. All the members of group I had introns in the range of 0–4. Similarly, for group II members, the number of introns ranged from one to six. Interestingly, group III had only two members, both of which contained 18 introns. Furthermore, the majority of the members of group IV had introns in the range of 0–1, whereas other members contained higher numbers of introns, even up to 10 or 11. Similarly, for group V, the majority of members were intronless; however, *VvBTB60, VvBTB68* and *VvBTB9* had two, two and one introns, respectively. Further motif analysis predicted that the motifs were scattered throughout the protein sequences, as shown in Figure 3. Some motifs were conserved within the members of the same group. For instance, motif 5 was conserved in almost all the members of group I. In addition, motif 3 or motif 5 were present in group II members. Interestingly, group III contained only two BTB proteins, in which motif 16 was conserved. Motif 11 was present in the majority of the members of group V. The details of different motifs are provided in Table S2.

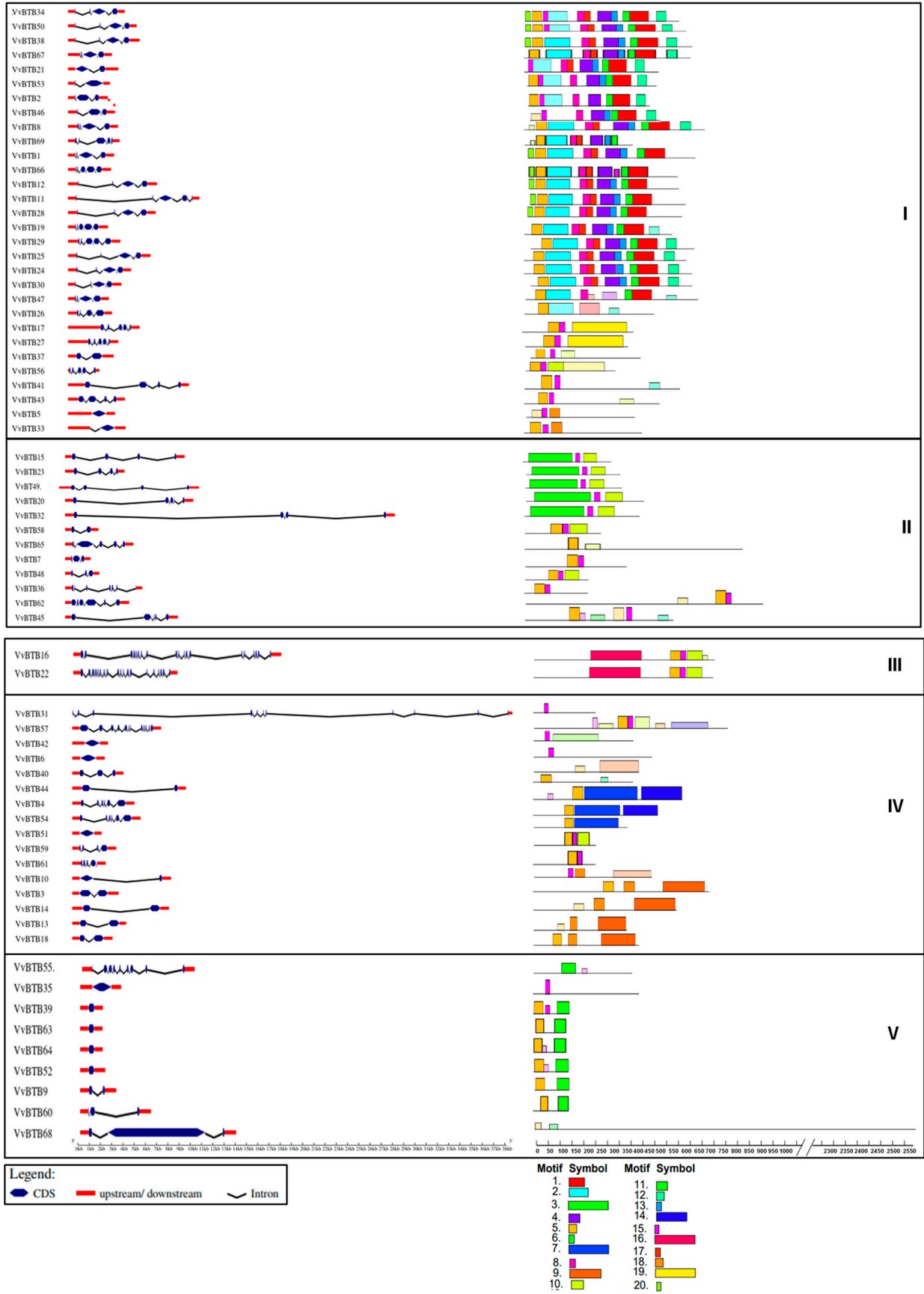

**Figure 3.** Groupwise intron–exon organizations and motif analysis. Conserved motif analysis using MEME and 20 different motifs are depicted by different-colored boxes.

Detailed domain analysis was also performed for all BTB proteins, and different colors of BTB domains exhibited different extensions at the C-terminal that were present along with the BTB domain (Figure S1). Moreover, these extension regions were crucial for protein–protein interactions and differed according to the presence of other adjacent domains. Our investigation of these conserved domains was based on the NCBI-CDD and InterProScan databases (Table S3).

### 3.4. Physicochemical Properties

BTB proteins were further explored for their physicochemical properties, including the number of amino acids, molecular weight, isoelectric point (pI), instability index and aliphatic index (Table 1). The average protein length, molecular weight, pI, aliphatic index and instability index of group I members were 563 aa, 63 kDa, 7.05, 91.49 and 47.6, respectively. For group II members, the average protein length, molecular weight, pI, aliphatic index and instability index were 469, 52 kDa, 6.01, 89.28 and 42.7, respectively. In addition, group III members exhibited an average amino acid length of 709 aa, with a 78 kDa molecular weight. The pI, aliphatic index and instability index averaged 6.08, 105.1 and 43.84, respectively. Similarly, 480, 53 kDa, 6.28, 88 and 50.39 were the average protein length, molecular weight, pI, aliphatic index and instability index for group IV, respectively. Group V members showed an average protein length of 487 aa and an average molecular weight of 25 kDa. The average pI, aliphatic index and instability index for group V members were 4.87, 79.1 and 38.11, respectively.

### 3.5. Subcellular Location

Most of the members of group I of BTB proteins were predicted to be localized in the nucleus based on two programs: WoLF PSORT and Cello v.2.5 (Table 1). Interestingly, some members, including *VvBTB43*, *VvBTB19*, *VvBTB8*, *VvBTB28* and *VvBTB34*, also showed dual localization in the cytoplasm and nucleus. However, the majority of group II members were present in the cytoplasm, except for *VvBTB58* and *VvBTB62*, which were present in the nucleus. Group III proteins showed their presence in the cytoplasm. Group IV members were found to be localized in the nucleus, cytoplasm, plasma membrane and/or the chloroplast. The majority of members of group V also exhibited their presence either in the nucleus or cytoplasm.

### 3.6. Structural Analysis of BTB Domain

The structural analysis of the BTB domain was achieved by generating a tertiary structure model through I-TASSER (Figure 4). The model generated through I-TASSER was also investigated on the basis of the C-score, which was 0.78 in our case. Usually, C-score values range from (−5 to 2), which means our model was a high-confidence model. Further, TM and RMSD values were $0.82 \pm 0.08$ and $2.5 \pm 1.9$, respectively, which signified a preferable prediction for the created model. Further, PDB data of the tertiary structure derived from I-TASSER was subjected to the ENDscript 2.0 server to view different secondary structures, such as alpha helices and beta-sheets of the BTB domain. A total of five alpha helices, with A1/2 and A4/5 forming hairpins of alpha helices, and three beta sheets were displayed in the BTB domain.

**Table 1.** Physicochemical properties and sub-cellular localization of *BTB* family in *V.vinifera* (AA: amino acid; MW: molecular weight; pI: isoelectric point).

| Gene IDs | Nomenclature | AA | MW (KDa) | pI | Instability Index | Aliphatic Index | Subcellular Location (CELLO) | Subcellular Location (WoLF PSORT) |
|---|---|---|---|---|---|---|---|---|
| VIT_19s0085g00180.t01 | VvBTB63 | 152 | 172.6 | 4.77 | 29.6 | 87.2 | cytoplasmic | cytoplasmic |
| VIT_03s0038g02500.t01 | VvBTB9 | 155 | 175.2 | 4.57 | 43.8 | 88.1 | cytoplasmic | cytoplasmic |
| VIT_19s0085g00190.t01 | VvBTB64 | 157 | 175.2 | 4.75 | 29.1 | 91.9 | cytoplasmic | cytoplasmic |
| VIT_15s0045g00490.t01 | VvBTB52 | 151 | 175.2 | 4.66 | 36 | 95.6 | cytoplasmic | nuclear |
| VIT_10s0003g05820.t01 | VvBTB39 | 162 | 189 | 4.77 | 58.1 | 93.3 | cytoplasmic | cytoplasmic |
| VIT_18s0001g14860.t01 | VvBTB60 | 197 | 225.2 | 4.97 | 49.9 | 89 | cytoplasmic | chloroplastic |
| VIT_08s0007g03590.t01 | VvBTB36 | 250 | 289.3 | 5.89 | 38.8 | 98.7 | plasma membrane | cytoplasmic |
| VIT_18s0001g01590.t01 | VvBTB58 | 264 | 298.4 | 5.17 | 60.7 | 83.2 | nuclear | chloroplastic |
| VIT_13s0067g02290.t01 | VvBTB48 | 270 | 299.3 | 5.75 | 41.1 | 85.3 | cytoplasmic | chloroplastic |
| VIT_19s0015g01430.t01 | VvBTB61 | 283 | 319.4 | 5.65 | 43.9 | 94.4 | chloroplast | cytoplasmic |
| VIT_08s0056g01620.t01 | VvBTB31 | 302 | 332.2 | 6.53 | 36 | 88.6 | extracellular/nuclear | chloroplastic |
| VIT_18s0001g02680.t01 | VvBTB59 | 326 | 374.7 | 6.03 | 51.7 | 90.7 | cytoplasmic | cytoplasmic |
| VIT_17s0000g09790.t01 | VvBTB56 | 347 | 398.6 | 9.11 | 60.3 | 87.3 | nuclear | nuclear |
| VIT_12s0028g00580.t01 | VvBTB44 | 375 | 416.9 | 7.62 | 51.9 | 94.5 | plasma membrane | chloroplastic |
| VIT_02s0012g00960.t01 | VvBTB7 | 382 | 417.6 | 5.58 | 52.2 | 102.1 | extracellular/cytoplasmic | chloroplastic |
| VIT_07s0129g00210.t01 | VvBTB27 | 371 | 419.6 | 9.22 | 42.7 | 94.4 | nuclear | chloroplastic |
| VIT_13s0019g02260.t01 | VvBTB49 | 406 | 445.3 | 6.17 | 38.6 | 87.9 | plasma membrane | cytoplasmic |
| VIT_07s0104g00570.t01 | VvBTB23 | 402 | 445.3 | 6.4 | 27 | 86.3 | cytoplasmic | chloroplastic |
| VIT_15s0046g02450.t01 | VvBTB55 | 389 | 449.7 | 5.26 | 49.8 | 82.7 | nuclear | nuclear |
| VIT_05s0020g02520.t01 | VvBTB15 | 408 | 454.3 | 6.64 | 29 | 84.6 | chloroplastic | chloroplastic |
| VIT_05s0094g00950.t01 | VvBTB17 | 407 | 455 | 9.02 | 49.4 | 87.4 | nuclear | nuclear |
| VIT_06s0009g01030.t01 | VvBTB20 | 423 | 464.8 | 6.08 | 42.3 | 78.6 | cytoplasmic | cytoplasmic |
| VIT_10s0042g00560.t01 | VvBTB40 | 420 | 467 | 6.41 | 41.9 | 97.1 | plasma membrane | plasma membrane |
| VIT_08s0056g01670.t01 | VvBTB32 | 431 | 475.8 | 5.47 | 40.7 | 80.8 | cytoplasmic/chloroplast | chloroplastic |
| VIT_14s0068g01350.t01 | VvBTB51 | 427 | 480 | 5.45 | 52 | 83.2 | nuclear | nuclear |
| VIT_02s0025g02270.t01 | VvBTB5 | 431 | 483.4 | 4.8 | 51.2 | 102.9 | cytoplasmic | cytoplasmic |
| VIT_11s0016g00110.t01 | VvBTB42 | 441 | 494.5 | 5.47 | 40.2 | 74.9 | cytoplasmic | cytoplasmic |
| VIT_08s0105g00220.t01 | VvBTB33 | 440 | 496.5 | 4.69 | 46 | 98.8 | plasma membrane/cytoplasmic | cytoplasmic |
| VIT_02s0025g03300.t01 | VvBTB6 | 464 | 504.5 | 6.98 | 42 | 86.7 | chloroplastic | cuclear |
| VIT_08s0007g01990.t01 | VvBTB35 | 475 | 515.4 | 5.26 | 43.7 | 78 | chloroplastic | cytoskeleton |
| VIT_08s0007g05740.t01 | VvBTB37 | 490 | 532.9 | 6.1 | 47.5 | 92.6 | nuclear | chloroplastic |
| VIT_03s0063g02520.t01 | VvBTB10 | 488 | 541.8 | 5.58 | 56.9 | 93.9 | nuclear | nuclear |
| VIT_06s0004g03710.t01 | VvBTB18 | 485 | 554.2 | 5.59 | 59.2 | 92.5 | nuclear | nuclear |
| VIT_07s0129g00070.t01 | VvBTB26 | 507 | 556.4 | 5.88 | 48.5 | 87.5 | nuclear | nuclear |

**Table 1.** *Cont.*

| Gene IDs | Nomenclature | AA | MW (KDa) | pI | Instability Index | Aliphatic Index | Subcellular Location (CELLO) | Subcellular Location (WoLF PSORT) |
|---|---|---|---|---|---|---|---|---|
| VIT_00s0665g00050.t01 | VvBTB69 | 501 | 566.9 | 5.57 | 52.7 | 93.8 | nuclear | nuclear |
| VIT_04s0023g01000.t01 | VvBTB13 | 524 | 590.1 | 6.65 | 61 | 90.9 | nuclear | cytoplasmic |
| VIT_01s0010g01390.t01 | VvBTB2 | 538 | 603 | 8.82 | 50 | 92 | nuclear | nuclear |
| VIT_12s0134g00010.t01 | VvBTB46 | 544 | 607.99 | 8.81 | 52.5 | 89 | plasma membrane/cytoplasmic/nuclear | nuclear |
| VIT_04s0044g01240.t01 | VvBTB14 | 551 | 621.8 | 8.28 | 57.3 | 86.3 | nuclear | nuclear |
| VIT_15s0046g01510.t01 | VvBTB54 | 553 | 623 | 5.33 | 52 | 79 | nuclear | nuclear |
| VIT_15s0048g02810.t01 | VvBTB53 | 559 | 630.6 | 5.62 | 55 | 89.9 | nuclear | chloroplastic |
| VIT_02s0025g00880.t01 | VvBTB4 | 562 | 634.4 | 5.36 | 51.4 | 79.7 | cytoplasmic/nuclear | nuclear |
| VIT_06s0004g08230.t01 | VvBTB19 | 572 | 637.1 | 6.88 | 39 | 93 | cytoplasmic/nuclear | chloroplastic |
| VIT_12s0059g00550.t01 | VvBTB45 | 574 | 648.4 | 5.08 | 47.6 | 88.9 | cytoplasmic | nuclear |
| VIT_11s0016g01990.t01 | VvBTB43 | 584 | 649.4 | 5.66 | 40.3 | 95 | nuclear | nuclear |
| VIT_06s0061g01140.t01 | VvBTB21 | 580 | 650.2 | 7.56 | 46.5 | 91.3 | nuclear | nuclear |
| VIT_10s0042g01250.t01 | VvBTB41 | 587 | 655.3 | 6.25 | 46.5 | 89.5 | nuclear | nuclear |
| VIT_08s0056g00610.t01 | VvBTB30 | 602 | 669.6 | 6.19 | 45.8 | 92.8 | cytoplasmic | chloroplastic |
| VIT_00s0194g00080.t01 | VvBTB66 | 593 | 673 | 8.73 | 50.4 | 86.8 | nuclear | cytoplasmic |
| VIT_13s0019g04420.t01 | VvBTB50 | 619 | 688.9 | 5.46 | 43.8 | 98.3 | nuclear | nuclear |
| VIT_03s0091g00680.t01 | VvBTB11 | 624 | 689.5 | 7.11 | 37.1 | 90.6 | cytoplasmic | cytoplasmic |
| VIT_04s0008g06630.t01 | VvBTB12 | 620 | 694.6 | 9.02 | 42 | 90.7 | nuclear | nuclear |
| VIT_07s0005g06300.t01 | VvBTB25 | 616 | 695.2 | 6.51 | 46.4 | 90.4 | nuclear | nuclear |
| VIT_07s0129g00250.t01 | VvBTB28 | 629 | 697 | 6.37 | 43 | 87.3 | cytoplasmic | cytoplasmic |
| VIT_08s0040g01800.t01 | VvBTB34 | 624 | 699.4 | 5.65 | 42.6 | 91.2 | cytoplasmic | cytoplasmic |
| VIT_10s0003g04490.t01 | VvBTB38 | 632 | 699.5 | 6.16 | 47.1 | 87.9 | cytoplasmic/nuclear | cytoplasmic |
| VIT_01s0137g00820.t01 | VvBTB1 | 630 | 700.8 | 8.78 | 55.2 | 95.4 | nuclear | nuclear |
| VIT_07s0104g01440.t01 | VvBTB24 | 631 | 704.8 | 8.95 | 48.4 | 94.4 | nuclear | chloroplastic |
| VIT_00s0207g00230.t01 | VvBTB67 | 635 | 707.4 | 5.12 | 54.3 | 87.3 | cytoplasmic/nuclear | chloroplastic |
| VIT_07s0129g00380.t01 | VvBTB29 | 636 | 709.6 | 6.47 | 44.9 | 94.8 | cytoplasmic/nuclear | nuclear |
| VIT_02s0025g00150.t01 | VvBTB3 | 667 | 737.965 | 7.96 | 65.3 | 87.3 | nuclear | chloroplastic |
| VIT_03s0038g00270.t01 | VvBTB8 | 674 | 757.194 | 8.15 | 42.9 | 82.6 | nuclear | nuclear |
| VIT_12s0057g00510.t01 | VvBTB47 | 674 | 763.1 | 8.97 | 55.8 | 89.4 | nuclear | chloroplastic |
| VIT_07s0104g00020.t01 | VvBTB22 | 705 | 780.7 | 6.22 | 44.3 | 108.7 | cytoplasmic | nuclear |
| VIT_05s0020g03020.t01 | VvBTB16 | 713 | 783.1 | 5.94 | 43.4 | 101.7 | cytoplasmic | cytoplasmic |
| VIT_18s0122g01340.t01 | VvBTB57 | 806 | 923.1 | 5.72 | 43.5 | 88.4 | plasma membrane | plasma membrane |
| VIT_00s0179g00100.t01 | VvBTB65 | 886 | 1001.8 | 5.52 | 50.4 | 93.6 | cytoplasmic | nuclear |
| VIT_19s0027g00550.t01 | VvBTB62 | 929 | 1048.2 | 8.4 | 44.6 | 101.3 | plasma membrane | nuclear |
| VIT_00s0349g00030.t01 | VvBTB68 | 2549 | undefined | undefined | 2.9 | 6 | extracellular/nuclear | cytoplasmic |

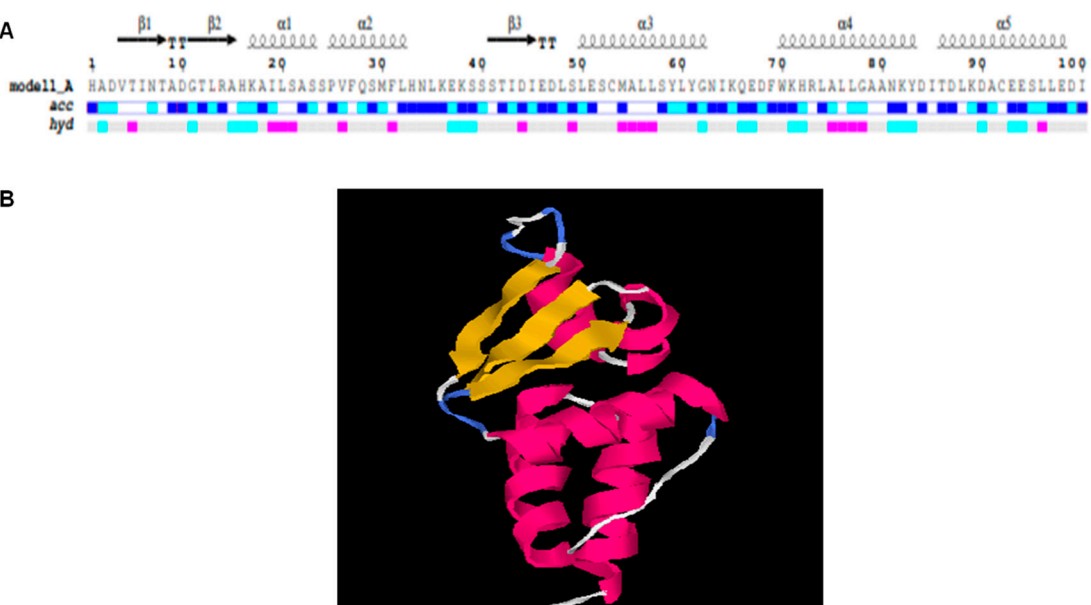

**Figure 4.** Structural analysis of BTB domain. (**A**) Secondary conformation of BTB domain represented in black-colored arrows (β-sheets: β1, β2, β3) and spiral forms (α-helices: α1, α2, α3, α4 and α5) along with sequence. (**B**) Detailed structural analysis indicates yellow-colored β-sheets and pink spiral forms are α-helices.

*3.7. Expression Analysis of BTB Gene Family*

The identification of differentially expressed *BTB* genes across various biotic (PM and DM) and abiotic (cold, heat and drought) stresses and three tissues (inflorescence, berry and leaf) at 10 developmental stages was achieved by determining FPKM values using the Trinity package. A HCL tree was generated to exhibit the relative expression of *VvBTB* genes at different developmental stages and stress conditions. In the case of biotic stress, 16 *BTB* genes exhibited differential expression in response to PM. Of these, five *VvBTB* genes were also responsive towards downy mildew (DM) infection, as shown in Figure 5A,B. In abiotic stress, the highest number of differentially expressed *BTB* genes was observed in heat (36 genes), followed by cold (23 genes) and drought (8 genes), respectively, as shown in (Figure 5C–E). Interestingly, 15 genes were common in heat and cold, 5 in heat and drought and 1 in cold and drought stress. Overall, 1 DEG was common to all abiotic stresses, i.e., *VvBTB27*. Similarly, 50, 49 and 44 genes were differentially expressed in inflorescence, leaf and berry, respectively. Additionally, all DEGs of leaf and berry were also expressed in inflorescence. Moreover, 44 *BTB* genes were commonly expressed in all the three tissues. Overall, 50 *BTB* genes were differentially expressed in response to tissue development, 47 in abiotic and 16 in biotic stress. However, 37 genes were responsive towards both development and abiotic stress with 14 towards abiotic and biotic and 14 towards biotic and developmental stages. Interestingly, 13 DEGs were commonly expressed in all three conditions.

Of all conditions, the maximum *BTB* genes (approx. 70%) were responsive in inflorescence development, whereas the minimum genes (approx. 7%) were responsive in DM. In addition, the maximum fold change value was shown in the case of developmental stages, i.e., +9 to −15, and the minimum was observed in biotic stress conditions, i.e., −2 to + 2, which also supports the plausible roles of *BTB* genes in developmental patterns of *Vitis.*

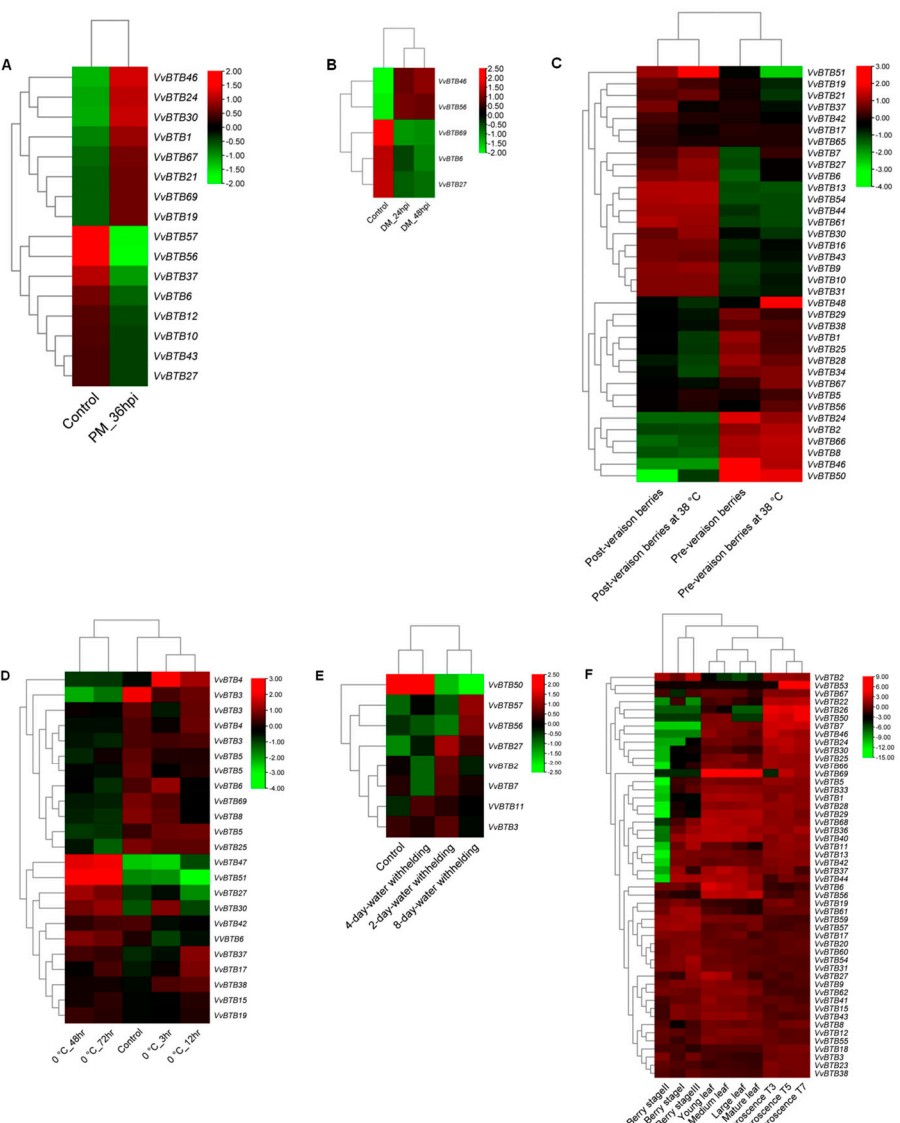

**Figure 5.** Heat maps depicting expression profiles of *VvBTB* genes in (**A**) PM infection (powdery mildew infection at 36 hpi), (**B**) DM (downy mildew infection at 24 and 48 hpi), (**C**) heat stress, (**D**) cold stress, (**E**) drought and (**F**) developmental stages. Heat maps were generated using TBtools. Color bars for each heat map indicate relative expression. Red color represents upregulation, black color represents no change in expression, and green color represents downregulation.

### 3.8. Prediction of Cis-Regulatory Element

To investigate the transcriptional regulation of 13 common differentially expressed *BTB* genes, the *cis*-regulatory elements present in their promoters were predicted using the PlantCARE database (Figure 6). After analysis of a 1.5 kb upstream region of *VvBTB* genes, results showed the presence of core *cis*-elements, including the CAAT-box and TATA box. These genes exhibited the presence of *cis*-elements in all categories, i.e., development-responsive elements, hormone-responsive elements and abiotic and biotic stress-responsive elements. Major development-responsive elements included AC-I (xylem-specific expression), CAT-box (responsive to meristem expression), Box-III (element encode for protein-binding site), O2-site (regulatory effect on zein metabolism), RY-element (responsive towards seed development), GCN4-motif (related to endosperm expression), 3-AF3 (conserved DNA module array), A-box (conserved for alpha amylase promoters), F-box (cell cycle regulation), C-TAG motif (development responsiveness), AACA-motif (endosperm-specific negative expression) and CCGTCC-motif for meristem-specific activa-

tion. Similarly, hormone-responsive elements such as TCA-element, Aux-RR core element and TGA-element showed auxin responsiveness, and TGACG-motif and CGTCA-motif were responsive towards methyl jasmonate. Further, gibberellin-responsive elements included P-box and GARE, and ABRE and ERE elements were associated with abscisic acid and ethylene responsiveness, which were present in all the selected *BTB* genes. Biotic stress-related elements included W-box, WUN-motif and WRE3, and abiotic stress-related elements included STRE (stress-responsive element), oxidative stress-responsive elements, i.e., as-1 and MYB, MYC, MBS1 (flavonoid biosynthetic gene regulation), various light-responsive elements, namely, Box4, TCCC motif, GT1-motif, Box II, G-box, L-box, Gap-box, TCT motif, ACTC-motif, AE-box, GATA motif, chs-CMA1a, sp-1, ACT motif, I-box, MRE (Myb-related element), ACE, GA-motif, 3-AF1 and ATC motif, drought-inducible element (MBS), LTR (low temperature-responsive element), DRE (drought-responsive element), CCAAT box (acting as a MYBHv1-binding site), GC-motif (anoxic-specific inducibility), defense- and stress-responsive TC-rich repeats and ARE (inducible in anaerobic conditions). Overall, the presence of diverse *cis*-elements supported the in silico expression analysis of *BTB* genes, which showed that 13 genes were responsive towards all conditions, i.e., developmental patterns and abiotic and biotic stresses.

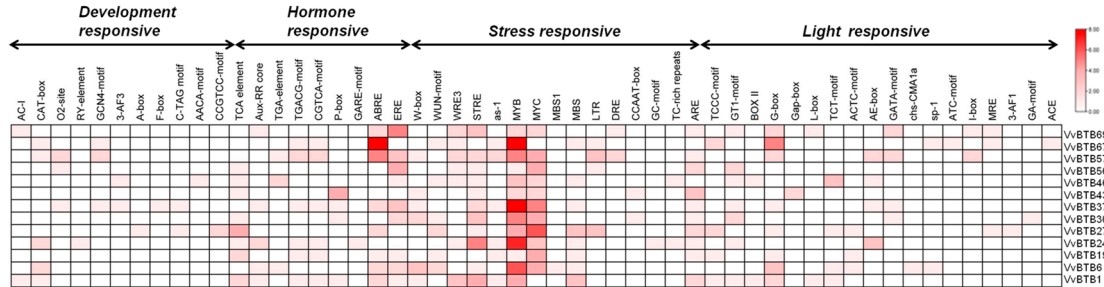

**Figure 6.** *Cis*-regulatory element analysis of the promoters of selected *BTB* genes. The color bar depicts the number of *cis*-regulatory elements.

*3.9. BTB Genes as miRNA Targets*

Interactions between miRNAs and *BTB* genes were predicted using the psRNATarget tool (Figure 7). A total of 186 published miRNAs specific to *V. vinifera* available in psRNATarget were used for the target analysis of 13 important DEGs. Our analysis revealed that six *VvBTB* genes acted as targets of diverse miRNAs. However, few *BTB* genes acted as targets of multiple miRNAs. For example, *VvBTB24* was targeted by vvi-miR172, vvi-miR169, vvi-miR399 and vvi-miR3623-5p and vvi-miR3625-5p miRNAs (Figure 7A). Similarly, *VvBTB1*, *VvBTB57* and *VvBTB6* were also predicted as targets of multiple miR-NAs (Figure 7B–D). In addition, vvi-miR482 and vvi-miR3636-5p targeted *VvBTB56* and *VvBTB30*, respectively (Figure 7E,F).

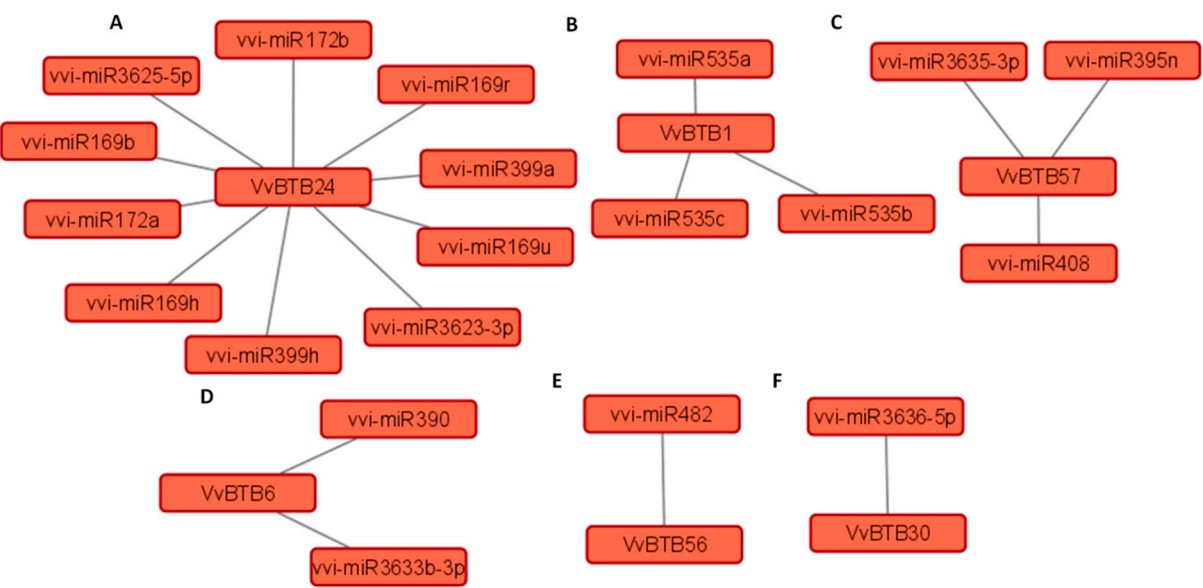

**Figure 7.** Interaction analysis of *BTB* genes, including (**A**) *VvBTB24*, (**B**) *VvBTB1*, (**C**) *VvBTB57*, (**D**) *VvBTB6*, (**E**) *VvBTB56* and (**F**) *VvBTB30* with *V. vinifera* miRNAs. Each interaction network was generated using Cytoscape 3.9 software.

## 4. Discussion

According to some recent reports, the BTB domain plays plausible roles in developmental and stress-related pathways in several plants [6,7,12,13,20–22,24,26,54,55]. Moreover, the availability of significant genomic and transcriptomic data of wine grapes encouraged us to attempt the genome-wide identification and characterization of the recently identified BTB domain containing a protein family. We investigated the involvement of *BTB* genes in developmental stages (10) of three different (leaf, berry and inflorescence) tissues and abiotic (cold, heat and drought) and biotic stresses (PM and DM). Previously, several *BTB* genes were identified in the genomes of various plants, namely, 158 genes in rice, 38 in tomato, 80 in *Arabidopsis* and 49 in sugar beet [10–13]. In line with previous reports, our study identified a total of 69 *BTB* genes in the *V. vinifera* genome. However, these differences in the number of *BTB* genes among diverse plant species may be due to genome size and long-term evolution. This evolutionary gene loss is a direct reflection of the biological importance of a particular gene family [2,56]. Further, we performed mapping of all the identified *BTB* genes on the *Vitis* genome. We observed a broad allocation of *VvBTB* genes in the plant genome. Interestingly, five *BTB* genes were mapped on an uncharacterized chromosome, and three were mapped at random positions of particular chromosomes. Random genes were assigned a chromosome number but without any certain placement within the chromosome [57]. Likewise, two members of the *SCPL* gene family were mapped on random positions of chromosome 7 and 10 [58]. In the present study, *VvBTB8, VvBTB42* and *VvBTB44* were found to be located at random positions on chromosome 3, 11 and 12, respectively. Based on the phylogenetic analysis, VvBTB proteins were clustered into five major groups. Interestingly, BTB proteins present within a group exhibited similarity in terms of protein size and intron–exon organization. Similarly, in rice, five groups of OsBTB proteins were identified that showed similarity in protein size and intron–exon organization within the group [11,12]. In addition, intron–exon organization revealed the presence of one to five introns in most of the *VvBTB* genes; however, some members contained a higher number of introns (10, 11, 18), whereas others were intronless. Our results agreed with a previous report in which it was concluded that tomato *BTB* genes also had introns ranging from 0–18. Additionally, motifs related to the BTB domain were also present in all the members of the BTB family of grapes, similar to previous studies [11]. NPH3-related motifs 1, 2, 4, 6, 8 and 13 were conserved among the members of group I,

suggesting the role of NPH3 in providing a specific function to these proteins. The majority of the group I genes were also differentially expressed during different developmental stages. These results are corroborated by a previous study in which a BTB-NPH3-like protein (NPY1) was shown to be involved in auxin-mediated organogenesis [59]. Similarly, motif 16 was conserved within the members of group III (VvBTB16 and VvBTB22). This motif is related the ABAP1/ARIA domain, which is known to play crucial role in ABA responses. One such protein, ARIA, also contains the BTB domain and interacts with ABF2 to regulate ABA-mediated responses in *Arabidopsis* [60]. We also conducted detailed domain analysis of each member of the VvBTB protein family. Our results showed that additional domains were present on the N- and C- terminal side of the BTB domain. As shown in Figure S1, the BTB_POZ_NPR_plant domain was present with various adjacent domains, namely, the DUF (Domain of Unknown Function), Ank repeats and the NPR1 domain at the C-terminal, and the BTB_POZ_BT domain was mostly present with the BACK and the Znf-TAZ domain at the C-terminal. Interestingly, it was also recently reported that the BTB domain plays a major role in the salicylic acid (SA)-signaling pathway by mediating SA receptors' (NPR1) oligomerization [6,7,61]. In addition, the BTB_POZ_BPM-plant domain was found with the MATH domain at the N-terminal and the BACK domain at the C-terminal, and BTB_POZ_ARIA-plant domain overlapped with the BACK_ARIA-like domain at the C-terminal. The BTB domain was present in all 69 members of the VvBTB family; however, it differs in its extension regions. These extensions are essential for protein–protein interactions occurring in the biological system [1,8]. For instance, the BTB_POZ domain present along with the NPH3 domain at the C-terminal exhibited the presence of extension NPY3 (BTB_POZ_ NPY3), and it was present in 22 members of group I. Furthermore, few members showed the presence of BTB domains with diverse extension regions such as the BTB_POZ_ZBTB_KHLH-like, BTB_POZ_ETO1-like, BTB_POZ_CP190-like, BTB1_POZ_ABTB1_BPOZ1, BTB_POZ_BTB, BACK_BTBD17, BTB_POZ_KCTD-like and BTB_POZ_FIP2-like domain, and these extensions were also previously reported in different plants such as *Arabidopsis*, tomato, rice and sugar beet [10–13]. There were two OsBTB and three SlBTB members that contained two BTB domains in rice and tomato, respectively [11,12]. Likewise, we also identified three VvBTB members, namely, VvBTB45, VvBTB62 and VvBTB57, which contained two BTB domains. Interestingly, one member of the tomato BTB family exhibited the presence of three BTB domains, but no VvBTB member was found to contain more than two BTB domains in *V. vinifera*. Our study agreed with a previous report in tomato in which the majority of BTB domains were found at the N-terminus, whereas some were present at the C-terminal [11]. Further physicochemical analysis of identified BTB proteins revealed a substantial variation in the protein length (151–929), molecular weight (17.2589–104.821 kDa) and isoelectric point (4.57–9.22). Isoelectric values clearly indicated that most of the BTB proteins were of acidic nature in *V. vinifera*. The instability index was greater than 40 for most of the members, indicating that most of the BTB proteins were unstable. The aliphatic index indicates the thermal stability of the protein, which was in the range of 79 to 105 for BTB proteins. A higher aliphatic index of BTB proteins indicated their high thermal stability. In sugar beet, most of the BTB proteins were localized in the nucleus; likewise, the majority of VvBTB proteins were predicted to be present in the nucleus. However, some members were located in the cytoplasm, chloroplast, plasma membrane or extracellular space, and some exhibited multiple sub-cellular locations [13]. In addition, structural analysis also exhibited that the BTB domain contained five alpha helices and three beta sheets, which play an important part in protein–protein interactions, and this was in correlation with previous reports of the BTB family of other plant species [11,13].

The expression analysis of *BTB* genes in different conditions determined the plausible roles of *BTB* genes in developmental stages, significantly (50 out of 69) in inflorescence (Figure 5). Previous reports also highlighted the role of *BTB* in development stages [16,20,21,43]. Similar to previous reports, *BTB* genes of grapevine also exhibited responsiveness towards abiotic and biotic stresses [11–13,22,24–26]. Very few reports of the

functional characterization of *BTB* genes are available, which indicate that their functional specificity is far from any final interpretation. However, according to recent reports, the BTB domain was identified as a mediator of oligomerization of NPR1, which act as a receptor of salicylic acid [6,7,61]. An *Arabidopsis* BTB protein (ATSIBP1) containing an additional Skp1 domain positively regulates salt tolerance by alleviating ROS accumulation [62]. The silencing of another BTB protein, CaBPM4, in capsicum resulted in enhanced salt and drought tolerance, and it exhibited reduced resistance against *Phytophthora capsci* [63]. Interestingly, expression analysis revealed that 13 *BTB* genes exhibited differential expression under all conditions. Moreover, the presence of both development- and stress-related regulatory elements in the promoter sequences of 13 selected *BTB* genes supported their differential expression under all conditions. Several hormone-responsive *cis*-elements were also present in the promoters of *VvBTB* genes. For example, the As-I element involved in auxin and methyl jasmonate (MeJA) signaling was present in the promoters of several *VvBTB* genes [64]. Methyl jasmonate (MeJA) plays a crucial role in plant biotic stress resistance. It was interesting to observe that MeJA-responsive elements such as TGACG and CGTCA were present in the promoters of only six *VvBTB* genes, namely, *VvBTB1*, *VvBTB19*, *VvBTB24*, *VvBTB37*, *VvBTB57* and *VvBTB67*. In addition, another MeJA-responsive element G-box was also present in eight *VvBTB* genes. These *cis*-elements were shown to drive the expression of the *VvPR1* gene in response to MeJA treatment [65]. Several stress-responsive *cis*-elements, such as TC-rich repeats, TCA element, GARE-motif, O2-site, ARE, MBS, MBS1 and LTR, were present in the promoter of several *VvBTB* genes. Similarly, these elements were identified in the promoter of a sweet potato BTB-TAZ gene, *IBT4*. Furthermore, its overexpression resulted in enhanced drought tolerance in *Arabidopsis* [22]. The GT1-motif was also present in the promoters of seven *VvBTB* genes. This motif was shown to induce the expression of *VvNAC36* TF in response to powdery mildew infection, suggesting the potential role of *VvBTB* genes in PM disease resistance [66]. Several light-responsive elements, including GT1-motif, AE-box, 3-AF1, ATC-motif, GA-motif, G-box, GATA-motif and MRE, were also identified [67]. Additionally, several development-responsive *cis*-elements were present in the promoter of some *VvBTB* genes, such as *VvBTBT1, VvBTB6, VvBTB24, VvBTB27, VvBTB37, VvBTB57* and *VvBTB67,* highlighting the potential role of these genes in the regulation of growth and development. For instance, AC-I, CAT-box and CCGTCC are involved in meristem- and xylem-specific induction. Similarly, F-box and GCN-4 were shown to be involved in cell cycle regulation and circadian control [67]. Furthermore, interaction analysis of *BTB* genes with *V. vinifera*-specific miRNAs revealed that selected *BTB* genes were targeted by various development- and stress-related miRNA families, such as vvi-miR169, vvi-miR172, vvi-miR399, vvi-miR482, vvi-miR535, vvi-miR390, vvi-miR395 and vvi-miR408 (Figure 7) [68–79]. In our study, we performed in silico analysis, which suggested the potential role of *BTB* genes in the regulation of different developmental stages and biotic and abiotic stresses in *V. vinifera*. Further functional characterization of potential *BTB* genes can provide deep insight into the possible role of these genes, which can be exploited to develop better breeding and genetic-engineering approaches.

## 5. Conclusions

In the present study, a total of 69 *BTB* genes were identified that encode 69 BTB domain-containing proteins in the *Vitis* genome. Further, structural, functional and physicochemical analysis of the *VvBTB* family was conducted. The BTB domain is present along with other adjacent domains, mainly NPH3, BACK, MATH, DUF, NPR1, Skp1 and ANK. Expression analysis revealed the plausible role of *VvBTB* genes in developmental patterns, especially in inflorescence. Interestingly, about 18% of *BTB* genes were commonly responsive towards development and biotic and abiotic stresses. The presence of diverse transcriptional regulatory elements and interactions of *VvBTB* genes with development- and stress-related miRNAs further supports their plausible roles in the management of diverse mechanisms in grapevine. Overall, our findings highlight the essentiality of the *BTB* gene family,

which should probably be selected for further in-depth exploration of development- and stress-related mechanisms in grapes.

**Supplementary Materials:** The following supporting information can be downloaded at: https://www.mdpi.com/article/10.3390/agriculture13020252/s1, Figure S1: Detailed domain analysis of 69 BTB proteins in *V. vinifera*; Table S1: Detailed information of RNA-seq data retrieved from NCBI-SRA database; Table S2: Detailed information of conserved motifs identified in *BTB* gene family using MEME server; Table S3: Detailed information on the domains identified in the BTB proteins. Different colors represent the BTB domain with different extensions.

**Author Contributions:** K.S. conceived the idea, designed the experiments and analyzed the results. N.G. (Nandni Goyal) collected and analyzed the data. D.V. and N.G. (Naina Garewal) assisted in analyzing the data. N.G. (Nandni Goyal) and M.B. compiled the results. N.G. (Nandni Goyal) wrote the manuscript. K.S., N.G. (Nandni Goyal) and M.B. All authors have read and agreed to the published version of the manuscript.

**Funding:** The authors have not received any funding to carry out this work.

**Institutional Review Board Statement:** Not applicable.

**Informed Consent Statement:** Not applicable.

**Data Availability Statement:** The authors confirm that the data supporting the findings of this study are available within the article and its supplementary materials.

**Acknowledgments:** Nandni Goyal is thankful to CSIR for awarding her a junior research fellowship. The authors are thankful to the Department of Biotechnology, Panjab University, Chandigarh for providing facilities to carry out the work.

**Conflicts of Interest:** The authors have declared that no competing interests exist.

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
