# Peer review of "Genome-Wide Identification of BTB Domain-Containing Gene Family in Grapevine (Vitis vinifera L.)"

_agriculture, doi:10.3390/agriculture13020252_

Round 1
Reviewer 1 Report
This manuscript identified 69 BTB proteins in Vitis vinifera genome and explored the biological function in different growth processes, biotic and abiotic stresses. The study provides a comprehensive analysis of the research on BTB protein in grapes and other fruit crops. There are some comments for the author:
1 Line 158 and 159, “right” should be “left”
2 BTB, a highly conserved structural domain, is known to have genome-wide studies in species such as tomato, rice and sugarbeet. It is also interesting to see how the 69 BTB proteins in this paper relate to the evolution of BTB in other species.
3 The Groups to which VvBTBs belong should be indicated in Figure 3 for a clearer reading of the figure, similarly in Figure 2.
4 The protein motifs can be used for function prediction. Does the distribution of these five conserved motifs in different Groups of BTB proteins show a pattern? Is it possible to predict the function of BTB proteins and distinguish them from each other? Please describe briefly.
5 In the expression analysis of the BTB genes family, the sampling status of different developmental stages, biotic and abiotic stress treatments need to be reflected in Materials and methods.
6 In the cis-element analysis in Figure 6, perhaps a categorical representation of the different response elements can provide a more convenient understanding.
Author Response
Response to query 1: As per reviewer suggestion, we have made the suggested correction. Response to query 2: We have already described the evolution of 69 VvBTB proteins with respect to other plants in the discussion section in the lines, “Previously, several BTB genes................ biological importance of particular gene family [58]. Response to query 3: As per reviewer suggestion, we have made the suggested correction in figure 2 and 3. Response to query 4: We have performed motif analysis using 20 motifs with width (20-200) which revealed a pattern of conserved motifs in different groups. It is possible to predict the specific function of different group members of BTB proteins. We have also added some previous studies to strengthen our predictions in the discussion section. Response to query 5: As per reviewer suggestion, we have made the suggested changes in the materials and methods section: “2.6.Transcriptomic data collection and expression profiling of BTB genes”. Response to query 6: As per reviewer suggestion, we have categorised the different cis-elements in Figure 6.Reviewer 2 Report
The MS ‘Genome wide identification of BTB domain containing gene family in grapevine (Vitis vinifera L.)’ studied the BTB domain containing gene family in grapevine, which is suitable for publication in agriculture, but should have some major revisions.
1) In abstract, ‘Our findings revealed the plausible roles of BTB gene family in developmental stages, 50 VvBTB were responsive towards development.’ This sentence has grammar problem. ‘Interestingly, 13 VvBTB genes exhibited differential expression in all conditions.’ This sentence is not clear.
2) There are different ways to write ‘RNA-seq’. RNA seq or RNA-seq?
3) The order of appearance of Table S1-S3 in the text needs to be adjusted. The first appearance in the text should be Table S1.
4) In Figure 3, for the legend, there are two BTB_POZ and three NPH3, in different colors. What are the differences? Should be clear. And in my opinion, Figure S1 may be more useful than the motif analysis of Figure 3.
5) In table 1, for VvBTB67, ‘un’ should be ‘Un’.
6) Figure 5 has much more problems. First, the layout is not beautiful. And, for the heatmaps, some have repeats, but some not. in Fig A, PM_control has two repeats, while PM_36h has three. In my opinion, the heatmaps don’t need to show the repeats, and using average may be better. The abbreviations, like PM, DM, HTPRE, CD, DS, etc, should all be clearly stated, or the readers can not well understand it. For G, H, I, J, these results just showing gene number may be meaningless, can be deleted.
Author Response
Response to query 1: As per reviewer suggestion, we have made the suggested correction.
Response to query 2: As per reviewer suggestion, we have made the suggested modification.
Response to query 3: As per reviewer suggestion, we have made the suggested modification. We have named the tables according to their appearance in the text.
Response to query 4: We have modified the motif analysis part of Figure 3. The details of different motifs are provided in Table S2.
Response to query 5: As per reviewer suggestion, we have made the suggested correction.
Response to query 6: As per reviewer suggestion, we have made the suggested modifications.
Reviewer 3 Report
Goyal et al., Predicted 69 BTB genes of V. vinifera and investigated the involvement of some of them in developmental stages of 3 different (leaf, berry and inflorescence) tissues, abiotic (cold, heat and drought) and biotic stresses (PM and DM). Although their analysis strategies are adequate and show some interesting results, some improvement is required for their manuscript to be accepted for publication.
Introduction
On lines 55-57 it is stated that "Furthermore, function of BTB genes remains unexplored in grapevine which is both commercially and economically important fruit crop engaging our focus on genome wide investigation of BTB gene family in this plant". While this is true and may have been your motivation for using this species, I suggest to indicate some biological/phylogenetic arguments that justify the use of Vitis vinifera as a study model.
Materials and methods
En general it is important to improve the description of the thresholds and parameters used for each analysis. There was virtually no indication of the criteria used to predict a result as positive or negative.
Results
1. The relevance of "Genome wide identification and chromosomal distribution" is not very clear to me. Was there any expected? Why? Is it really important to include this figure if the information is repeated in Table 1?
2. Were BTB paralogous genes identified in the genome?
3. I suggest indicating the name of the groups (n=5) and also include a scale of sizes in Figure 3.
4. The design of table 1 should be improved. For example it would be better to use KDa rather than Da in the MW. Moreover, sort the table by MW or Subcellular Location could be more informative than sorting them by chromosomal location.
5. How was the quality of the data evaluated for RNA-Seq analysis? How many relevant experiments are available in the SRA database and how many were entered into your study? What were the inclusion/exclusion criteria?
6. For the analyses shown in Figure 5, I suggest performing Hierarchical Clustering for both the genes and the conditions evaluated. This will allow to obtain more and better information from the data used.
7. The amount of information obtained for the analysis of "Prediction of cis-regulatory element" appears to be very large. Is it possible to increase the requirements to obtain more reliable "cis-regulatory element"? Or do you have evidence that each of these genes could be as broadly regulated as your predictions suggest?
Discussion
1. In general, I suggest deepening the discussion of the main results of the study and highlighting the limitations that these could have considering that all of them are bioinformatic predictions.
2. In particular, I suggest to emphasize in defining which results were expected according to what has been described in other plant species and which results represent a novelty in Vitis vinifera.
Author Response
Introduction
Response to query: As per reviewer suggestion, we have added the biological relevance to use Vitis vinifera as a study model in the introduction section, “Being the perennial.................. of important gene families”.
Materials and methods
Response to query: We have used default parameters for some of the commonly used tools and have also given citation for this. We have also mentioned the required thresholds and parameters for other analysis for instance, “Subsequently, DEGs which displayed at least 1.5-fold................parameters”.
Results
Response to query 1: In Vitis vinifera, 69 BTB genes were broadly allocated on the chromosomes which are similar to rice BTB gene distribution. In contrast, most of the BTB genes were present on three chromosomes in tomato. We have also removed the chromosomal location data from the Table 1 because Figure 1 is useful in getting the visual representation of gene distribution.
Response to query 2: We have not identified the paralogous BTB genes in the genome.
Response to query 3: As per reviewer suggestion, we have indicated the group names and also added the scales of size in Figure 3.
Response to query 4: As per reviewer, we have included all the suggested changes in Table 1.
Response to query 5: We have downloaded the RNA-seq data from NCBI-SRA database (Table S1) and it’s quality was assessed as mentioned in the cited studies (Zhu et al, 2018; Hu et al, 2018; Bhatia et al, 2019). We have considered all the experiments available in the SRA database.
Response to query 6: As per reviewer suggestion, we have performed the hierarchical clustering for both the genes and conditions in Figure 5.
Response to query 7: As per reviewer suggestion, we have only included important cis-elements and grouped them into different categories. We have also added some previous studies in the support of our prediction, for instance, “several hormone responsive.........................................in cell cycle regulation and circadian control”.
Discussion
Response to query 1: As per reviewer suggestion, we have added several previous studies to support our in silico data, for instance, “NPH3 related motifs............................................ABA mediated responses in arabidopsis”, “several hormone responsive.........................................in cell cycle regulation and circadian control”.
Response to query 2: We have already described the results with respect to other plants in the discussion section. We have also mentioned the novelty in Vitis vinifera with respect to other plants, for instance, interestingly, one member............. domains in V. vinifera”, “Interestingly, 5 BTB genes................. the chromosome [59]”, Interestingly, expression analysis............. in all conditions”.
Round 2
Reviewer 2 Report
I think the author has made complete revision based on my suggestions, and the manuscript can be accepted in its current state.